# Concept Learning with Energy-Based Models

**Igor Mordatch**
OpenAI
San Francisco
`mordatch@openai.com`

## 1 Overview

Many hallmarks of human intelligence, such as generalizing from limited experience, abstract reasoning and planning, analogical reasoning, creative problem solving, and capacity for language and explanation are still lacking in the artificial intelligent agents. We, as others (Rosch et al. (1976); Lakoff & Johnson (1980); Lake et al. (2016)) believe what enables these abilities is the capacity to consolidate experience into *concepts*, which act as basic building blocks of understanding and reasoning.

Examples of concepts include visual (*"red"* or *"square"*), spatial (*"inside"*, *"on top of"*), temporal (*"slow"*, *"after"*), social (*"aggressive"*, *"helpful"*) among many others Lakoff & Johnson (1980). These concepts can be either identified or generated - one can not only find a square in the scene, but also create a square, either physical or imaginary. Importantly, humans also have a largely unique ability to combine concepts compositionally (*"red square"*) and recursively (*"move inside moving square"*) - abilities reflected in the human language. This allows expressing an exponentially large number of concepts, and acquisition of new concepts in terms of others. We believe the operations of identification, generation, composition over concepts are the tools with which intelligent agents can understand and communicate existing experiences and reason about new ones.

Crucially, these operations must be performed on the fly throughout the agent's execution, rather than merely being a static product of an offline training process. We believe execution-time optimization, similar to recent work on meta-learning (Finn et al. (2017)) plays a key role in this. We pose the problem of parsing experiences into a compositional, recursive arrangement of concepts as well as the problems of identifying and generating concepts as optimizations performed during execution lifetime of the agent. The meta-level training is performed by taking into account such processes in the inner level.

Specifically, a concept in our work is defined by an energy function taking as input an event configuration (represented as trajectories of entities in the current work), as well as an attention mask over entities in the event. Zero-energy event and attention configurations imply that event entities selected by the attention mask satisfy the concept. Compositions of concepts can then be created by simply summing energies of constituent concepts. Given a particular event, optimization can be used to identify entities belonging to a concept by solving for attention mask that leads to zero-energy configuration. Similarly, an example of a concept can be generated by optimizing for a zero-energy event configuration. See Figure 1 for examples of these two processes.

The energy function defines a family of concepts, from which a particular concept is selected with a specific concept code. Encoding of event and attention configurations can be achieved by execution-time optimization over concept codes. Once an event is encoded, the resulting concept code structure can be used to re-enact the event under different initial configurations (task of imitation learning), recognize similar events, or concisely communicate the nature of the event. We believe there is a strong link between concept codes and language, but leave it unexplored in this work.

At the meta level, the energy function is the only entity that needs to be learned. This is different from generative model or inverse reinforcement learning approaches, which typically also learn an explicit generator/policy function, whereas we define it implicitly via optimization. Our advantage as that the learned energy function can be transferred to other domains, for example using a robot to generate concepts in the physical world. Such transfer is not possible with an explicit generation/policy function, as it is domain-specific.

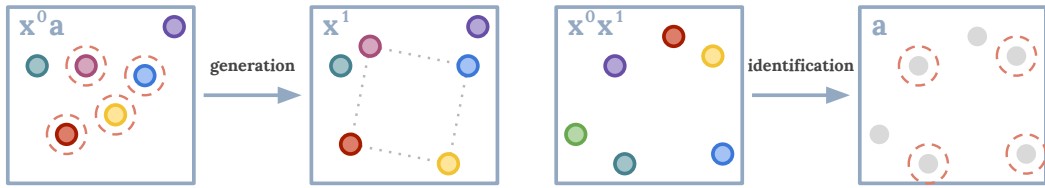

Figure 1: Examples of generation and identification processes for a *"square"* concept. a) Given initial state $\mathbf{x}^0$ and attention mask $\mathbf{a}$, square consisting of entities in $\mathbf{a}$ is formed via optimization over $\mathbf{x}^1$. b) Given states $\mathbf{x}$, entities comprising a square are found by optimization over attention mask $\mathbf{a}$.

## 2 ENERGY-BASED CONCEPT MODELS

Concepts operate over events, which in this work is a trajectory of $T$ states $\mathbf{x} = \left[\mathbf{x}^0, ..., \mathbf{x}^T\right]$. Each state contains a collection of $N$ entities $\mathbf{x}^t = [\mathbf{x}_0, ..., \mathbf{x}_N]$ and each entity $\mathbf{x}_i^t$ can contain information such as position and color of the entity. Attention over entities in the event is specified by a mask $\mathbf{a} \in \mathbb{R}^N$ over each of the entities.

Existence of a particular concept is given by energy function $E(\mathbf{x}, \mathbf{a}, \mathbf{w}) \in \mathbb{R}^+$, where parameter vector $\mathbf{w}$ specifies a particular concept from a family. The interpretation of $\mathbf{w}$ is similar to that of a code in an autoencoder. $E(\mathbf{x}, \mathbf{a}, \mathbf{w}) = 0$ when state trajectory $\mathbf{x}$ under attention mask $\mathbf{a}$ over entities satisfies the concept $\mathbf{w}$. Otherwise, $E(\mathbf{x}, \mathbf{a}, \mathbf{w}) > 0$. The conditional probabilities of a particular event configuration belonging to a concept and a particular attention mask identifying a concept are given by the Boltzmann distributions:

$$p(\mathbf{x}|\mathbf{a}, \mathbf{w}) \propto \exp\left\{-E(\mathbf{x}, \mathbf{a}, \mathbf{w})\right\} \qquad p(\mathbf{a}|\mathbf{x}, \mathbf{w}) \propto \exp\left\{-E(\mathbf{x}, \mathbf{a}, \mathbf{w})\right\} \qquad (1)$$

Given concept code $\mathbf{w}$, the energy function can be used for both generation and identification of a concept implicitly via optimization (see Figure 1):

$$\mathbf{x}(\mathbf{a}) = \operatorname*{argmin}_{\mathbf{x}} E(\mathbf{x}, \mathbf{a}, \mathbf{w}) \qquad \mathbf{a}(\mathbf{x}) = \operatorname*{argmin}_{\mathbf{a}} E(\mathbf{x}, \mathbf{a}, \mathbf{w}) \qquad (2)$$

Samples from distributions in (1) can be generated via stochastic gradient Langevin dynamics (Welling & Teh (2011)), effectively performing stochastic minimization in (2):

$$\tilde{\mathbf{x}} \sim \pi_x \left( \cdot \mid \mathbf{a}, \mathbf{w} \right) = \mathbf{x}^K, \;\; \mathbf{x}^k = \mathbf{x}^{k-1} + \frac{\alpha}{2}\nabla_{\mathbf{x}}E(\mathbf{x}, \mathbf{a}, \mathbf{w}) + \omega^k$$

$$\tilde{\mathbf{a}} \sim \pi_a \left( \cdot \mid \mathbf{x}, \mathbf{w} \right) = \mathbf{a}^K, \;\; \mathbf{a}^k = \mathbf{a}^{k-1} + \frac{\alpha}{2}\nabla_{\mathbf{a}}E(\mathbf{x}, \mathbf{a}, \mathbf{w}) + \omega^k, \;\; \omega^k \sim \mathcal{N}(0, \alpha) \qquad (3)$$

This stochastic optimization procedure is performed during execution time of the algorithm and is reminiscent of the Monte Carlo sampling procedures in prior work on energy-based models (Hinton (2006); Salakhutdinov & Hinton (2009); Friston (2010)), which are not differentiable and present tractability issues for learning. However, in our case the sampling procedure is differentiable, which makes the meta-level learning problem tractable via end to end back-propagation. The above procedure also differs from approaches that use explicit generator functions (Dayan et al. (1995); Kingma & Welling (2013); Kim & Bengio (2016)) or explicit attention mechanisms, such as dot product attention (Olah & Carter (2016)). While explicitly learned functions may be faster to evaluate, they offer limited generalization and cannot be transferred across domains.

## 3 LEARNING CONCEPTS FROM EVENTS

To learn concepts from experience grounded in events, we pose a few-shot prediction task. Given a few demonstration examples $X^{\text{demo}}$ containing tuples $(\mathbf{x}, \mathbf{a})$ and initial state $\mathbf{x}^0$ for a new event in $X^{\text{train}}$, the task is to predict attention $\mathbf{a}$ and the future state trajectory $\mathbf{x}^{1:T}$ of the new event. The new event may contain a different configuration or number of entities, so it is not possible to directly transfer attention mask, for instance. To simplify notation, we consider prediction of only one future state $\mathbf{x}^1$, although predicting more states is straightforward. The procedure is depicted in Figure 2.

Figure 2: Example of a few-shot prediction task we use to learn concept energy functions.

We follow the maximum entropy inverse reinforcement learning formulation (Ziebart et al. (2008)) and assume demonstrations are samples from the distributions given by the energy function $E$. Given a concept code $\mathbf{w}$, finding energy function parameters $\theta$ is posed as as maximum likelihood estimation problem over future state and attention given initial state. The resulting loss for dataset $X$ is

$$\mathcal{L}_\theta^{\mathrm{ML}}(X, \mathbf{w}) = \mathbb{E}_{(\mathbf{x}, \mathbf{a}) \sim X} \left[ -\log p_\theta \left( \mathbf{x}^1, \mathbf{a} \mid \mathbf{x}^0, \mathbf{w} \right) \right] \tag{4}$$

Where the joint probability can be decomposed in terms of probabilities in (1) as

$$p_\theta \left( \mathbf{x}^1, \mathbf{a} \mid \mathbf{x}^0, \mathbf{w} \right) = p_\theta \left( \mathbf{x}^1 \mid \mathbf{a}, \mathbf{w}_x \right) p_\theta \left( \mathbf{a} \mid \mathbf{x}^0, \mathbf{w}_a \right), \quad \mathbf{w} = [\mathbf{w}_x, \mathbf{w}_a] \tag{5}$$

We use two concept codes, $\mathbf{w}_x$ and $\mathbf{w}_a$ to specify the joint probability. The interpretation is that $\mathbf{w}_x$ specifies the concept of the action that happens in the event (i.e. *"be in center of"*) while $\mathbf{w}_a$ specifies the argument the action happens over (i.e. *"square"*). This is a concept structure or syntax that describes the event. The same concept can be used either as action or as an argument (because the energy function defining the concept can either be used for generation or identification). This importantly allows concepts to be understood from their usage under multiple contexts.

Conditioned on the two codes concatenated as $\mathbf{w}$, the negative logarithm of joint likelihood in (5) can be approximated as (see Appendix for the derivation)

$$-\log p_\theta \left( \mathbf{x}^1, \mathbf{a} \mid \mathbf{x}^0, \mathbf{w} \right) \approx \left[ E_\theta(\mathbf{x}^1, \mathbf{a}, \mathbf{w}_x) - E_\theta(\tilde{\mathbf{x}}, \mathbf{a}, \mathbf{w}_x) \right]_+ + \left[ E_\theta(\mathbf{x}^0, \mathbf{a}, \mathbf{w}_a) - E_\theta(\mathbf{x}^0, \tilde{\mathbf{a}}, \mathbf{w}_a) \right]_+$$

$$\tilde{\mathbf{x}} \sim \pi_x \left( \cdot \mid \mathbf{a}, \mathbf{w}_x \right), \quad \tilde{\mathbf{a}} \sim \pi_a \left( \cdot \mid \mathbf{x}^0, \mathbf{w}_a \right)$$

Where $[\cdot]_+ = \log(1 + \exp(\cdot))$ is the softplus function. This form is similar to contrastive divergence (Hinton (2006)) and structured SVM forms (Belanger et al. (2017)) and is a special case of guided cost learning formulation (Finn et al. (2016)). The approximation comes from sample-based estimates of the partition functions for $p(\mathbf{x})$ and $p(\mathbf{a})$.

The above equation makes use of sampling distributions $\pi_x$ and $\pi_a$ to estimate the respective partition functions. The approximation error in these estimates is minimal when KL divergence between $\pi$ and $\exp\{-E\}/Z$ is minimized, which intuitively encourages sampling distributions to generate samples from low-energy regions and is expressed by the following loss

$$\mathcal{L}_\pi^{\mathrm{KL}}(X, \mathbf{w}) = \mathbb{E}_{(\mathbf{x}, \mathbf{a}) \sim X} \left[ E(\tilde{\mathbf{x}}, \mathbf{a}, \mathbf{w}_x) + E(\mathbf{x}^0, \tilde{\mathbf{a}}, \mathbf{w}_a) \right] + \mathrm{H}\left[ \pi_x \right] + \mathrm{H}\left[ \pi_a \right] \tag{6}$$

**Execution-Time Inference of Concepts**  Given a set of example events, the concept codes can be inferred at execution-time via maximum likelihood estimation over codes $\mathbf{w}$

$$\mathbf{w}_\theta^*(X) = \operatorname*{argmin}_{\mathbf{w}} \mathcal{L}_\theta^{\mathrm{ML}}(X, \mathbf{w}) \tag{7}$$

This minimization is similar to execution-time parameter adaptation and the inner update of meta-learning approaches (Finn et al. (2017)). We perform the optimization with stochastic gradient updates similar to equation (3). Note that this optimization is not over meta-level parameters $\theta$, but is instead over function inputs that are unique for every batch entry. See Appendix for more details.

## 4 CONCLUSION

We believe that execution-time optimization plays a crucial role in acquisition and generalization of knowledge, planning and abstract reasoning, and communication. In this preliminary work, we proposed energy-based concept models as basic building blocks over which such optimization procedures can fruitfully operate (See Appendix for a brief description of our evaluation). In the current work we used a simple concept structure, but more complex structure with multiple arguments or recursion would be interesting to investigate in the future.

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

APPENDIX

DERIVATION OF JOINT LOG-LIKELIHOOD APPROXIMATION

The derivation is similar to guided cost learning (Finn et al. (2016)) and cost learning in linearly-solvable MDP (Dvijotham & Todorov (2010)) formulations. Joint negative log-likelihood of observing tuple $\left(\mathbf{x}^0, \mathbf{x}^1, \mathbf{a}\right)$ is $-\log p\left(\mathbf{x}^1, \mathbf{a} \mid \mathbf{x}^0, \mathbf{w}\right)$

$$= -\log\left(p\left(\mathbf{x}^1 \mid \mathbf{a}, \mathbf{w}_x\right) p\left(\mathbf{a} \mid \mathbf{x}^0, \mathbf{w}_a\right)\right) \tag{8}$$

$$= -\log\frac{\exp\left\{-E(\mathbf{x}^1, \mathbf{a}, \mathbf{w}_x)\right\}}{\int_{\tilde{\mathbf{x}}}\exp\left\{-E(\tilde{\mathbf{x}}, \mathbf{a}, \mathbf{w}_x)\right\}} - \log\frac{\exp\left\{-E(\mathbf{x}^0, \mathbf{a}, \mathbf{w}_a)\right\}}{\int_{\tilde{\mathbf{a}}}\exp\left\{-E(\mathbf{x}^0, \tilde{\mathbf{a}}, \mathbf{w}_a)\right\}} \tag{9}$$

Consider a more general form of the two individual terms above with non-negative function $f$

$$-\log\frac{\exp\left\{-f(\mathbf{x})\right\}}{\int_{\tilde{\mathbf{x}}}\exp\left\{-f(\tilde{\mathbf{x}})\right\}} = f(\mathbf{x}) + \log\mathbb{E}_{\tilde{\mathbf{x}}\sim q}\left[\frac{\exp\left\{-f(\tilde{\mathbf{x}})\right\}}{q(\tilde{\mathbf{x}})}\right] \tag{10}$$

The equality follows due to importance sampling under distribution $q$. There are a number of choices for sampling distribution $q$, but a choice that simplifies the above expression and we found to give stable results in practice is $q(X) = \frac{1}{2}\mathbb{I}\left[X = \mathbf{x}\right] + \frac{1}{2}\mathbb{I}\left[X = \tilde{\mathbf{x}}\right]$ where $\tilde{\mathbf{x}}\sim\pi$ and $\pi$ is a distribution that minimizes $\mathrm{KL}\left(\pi(X)\|\exp\left\{-f(X)\right\}/Z\right)$.

In this case, sample-based approximation of equation (10) leads to

$$f(\mathbf{x}) + \log\mathbb{E}_{\tilde{\mathbf{x}}\sim q}\left[\frac{\exp\left\{-f(\tilde{\mathbf{x}})\right\}}{q(\tilde{\mathbf{x}})}\right] \approx f(\mathbf{x}) + \log\left(\exp\left\{-f(\mathbf{x})\right\} + \exp\left\{-f(\tilde{\mathbf{x}})\right\}\right) \tag{11}$$

$$= \log\left(1 + \exp\left\{f(\mathbf{x}) - f(\tilde{\mathbf{x}})\right\}\right) = \left[f(\mathbf{x}) - f(\tilde{\mathbf{x}})\right]_+ \tag{12}$$

Using the above approximation in equation (9), gives the desired result $-\log p\left(\mathbf{x}^1, \mathbf{a} \mid \mathbf{x}^0, \mathbf{w}\right)$

$$\approx \left[E(\mathbf{x}^1, \mathbf{a}, \mathbf{w}_x) - E(\tilde{\mathbf{x}}, \mathbf{a}, \mathbf{w}_x)\right]_+ + \left[E(\mathbf{x}^0, \mathbf{a}, \mathbf{w}_a) - E(\mathbf{x}^0, \tilde{\mathbf{a}}, \mathbf{w}_a)\right]_+ \tag{13}$$

$$\text{where } \tilde{\mathbf{x}}\sim\pi_x\left(\,\cdot\mid\mathbf{a}, \mathbf{w}_x\right), \ \ \tilde{\mathbf{a}}\sim\pi_a\left(\,\cdot\mid\mathbf{x}^0, \mathbf{w}_a\right)$$

META-LEVEL PARAMETER OPTIMIZATION

We seek parameters $\theta$ of the energy function $E$ that maximize the likelihood of training data $X^{\text{train}}$ given demonstration data $X^{\text{demo}}$. Additionally, we seek sampling distributions $\pi$ that generates samples corresponding to low energies. However, the sampling distribution is implicitly a function of energy parameters $\theta$ via equations (3), a dependence which we denote as $\pi(\theta)$. The meta-level parameters are found via the following optimization problem

$$\min_\theta \mathcal{L}_\theta^{\text{ML}}(X^{\text{train}}, \mathbf{w}_\theta^*(X^{\text{demo}})) + \mathcal{L}_{\pi(\theta)}^{\text{KL}}(X^{\text{train}}, \mathbf{w}_\theta^*(X^{\text{demo}}))$$

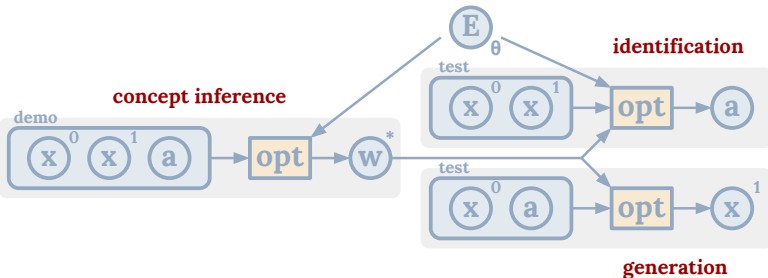

Figure 3: Execution-time inference in our method and the resulting nested optimization problems.

The difficulty is that even evaluation of $\mathcal{L}$ requires two nested levels of gradient-based optimization problems - first optimization level to infer $\mathbf{w}$ and second optimization level to generate samples $\tilde{\mathbf{x}}$ and $\tilde{\mathbf{a}}$ (see Figure 3). In order to backpropagate over such procedure, we turn nested optimizations into a sequence of alternating optimizations.

ENERGY FUNCTION DETAILS

There are many possible choices for the energy function as long as it is non-negative. The specific form we use in this work is

$$E_\theta(\mathbf{x}, \mathbf{a}, \mathbf{w}) = f_\theta(\sum_{t,i} \sigma(\mathbf{a}_i) \cdot g_\theta(\mathbf{x}_i^t, \mathbf{w}^g), \mathbf{w}^f)^2, \quad \mathbf{w} = [\mathbf{w}^f, \mathbf{w}^g]$$

Where $f$ and $g$ are multi-layer neural networks that each take concept code as part of their input. $\sigma$ is the sigmoid function and is used to gate the entities by the attention mask.

## 5 EXPERIMENTAL RESULTS

Due to space constraints, we are unable to describe our experiments in detail. Briefly, we use a two-dimensional particle-based environment to investigate execution-time concept generation, identification and inference and via optimization. We found success in spatial and relational concepts similar to examples shown in the figures. We have also investigated transfer of concept generation between domains, learning concept models in a the simple particle domain and then successfully enacting the concepts in a different environment with a physically simulated robot arm using model-predictive control (Tassa et al. (2012)).

