# OpenReview forum: "Concept Learning with Energy-Based Models"
_ICLR.cc/2018/Workshop — Accept_

### Official Review · AnonReviewer1 · 2018-03-10
**Relevant topic, with still very preliminary effort.**

**Rating:** 5
**Confidence:** 4

**Review:**

The author focuses on concept learning (where the emphasis is on composition of concepts to build new concepts) by an intriguing energy-based strategy, where each concept is encoded by an energy function, and composition is obtained by operations on these functions. This is a relevant topic, and some of the ideas on putting this together are original and welcome.

However, the paper is quite hard to follow, because some paragraphs are not clear. For instance, the first two paragraphs of Section 1 are very nice, but then the next paragraphs try to explain the main ideas with success. What is really the semantics behind these concepts? What is the intended effect of building new concepts out of arithmetic operations? There are some bold statements concerning how to transfer knowledge, but they do not seem justified.

Also, I find Figure 1 and related explanations to be quite poor. I could not really understand the main point of the attention mechanism, and could not find a clear definition of it (not a discussion of its intended purpose).

Note that "events" as discussed in this paper are often called "histories" in the MDP literature. It would be nice to be clear about the temporal aspects from the beginning. And then it is not clear that Expression (2) produces the right thing; is this an expression that yields an inference x? Or it is supposed to produce estimates of x (estimates do not seem to make sense here)? Also, it is not clear that p(x|a,w) is the "probability that x is in concept w"... what would be this probability?

Finally, the paper seems really preliminary in that there is no real test, no real example, and no real empirical validation here. Some additional focus on applications and on empirical testing seems necessary.

---

### Official Review · AnonReviewer2 · 2018-03-11
**Very interesting but without empirical evaluation while the motivation of the work is its online usage**

**Rating:** 7
**Confidence:** 2

**Review:**

The author investigate the important problem of concept consolidation (including recognition/identification and generation of instances of concepts). The approach is based around zero-energy models and execution-time optimization of these models attention (concept recognition/identification) and generation. The aim is to learn models of concepts which are transferable between domains as opposed to the traditional generative models and reinforcement learning approaches which are very domain specific.

The ideas and the method presented are interesting and appear to be consistent and well founded in the literature. The presented work is to the best of my knowledge novel. The problem under investigation is highly relevant and very important. The paper is further clearly written and of high quality.

The lack of inclusion of alleged experiments diminish the impact and reduces the possible comprehension of the strengths/drawbacks of the presented approach. Space constraints are blamed but there is at least almost a full page in the appendix that could be fully utilized for providing additional details and/or analysis/discussion of strengths/drawbacks.

I am not very familiar with much of the cited related work so it is hard for me to assess the significants of the contributions in this paper. I do however find the work interesting and would like to see a discussion between the author and others more familiar with the background material.

The solution appears elegant even though I have some trouble comprehending all the details and motivations. I am however skeptical (without any empirical evaluation) how feasible (and scalable) the approach is for online-usage, including the learning of all(?) parameters, (which is one of the mentioned motivations in the introduction).

Minor language errors exist throughout the paper. For example
"in a the simple" -> "in a simple"
"and via optimization" -> "via optimization" ?

---

### Official Review · AnonReviewer3 · 2018-03-12
**Energy based models for concept learning and identification**

**Rating:** 7
**Confidence:** 3

**Review:**

This paper suggests that learned energy functions combined with runtime optimization are sufficient to both recognise and generate concepts from small sample size data, through finding either a minimum energy attention (gives an example of the concept) or a minimum energy concept example (gives an instance of the attention), both conditioned on a fixed energy function parameterisation and concept code.

I think the paper is interesting but leaves out a lot of detail (due to space constraints).  Think it would make a good workshop presentation and poster.

You mention that the framework allows for the composition of concepts via straightforward composition of energies, I am curious as to how you envision recursion (the other template for structure generation you suggest is key to human intelligence) will be implemented?

You mention that energy-based models are building blocks over which such meta concept learning via optimization can fruitfully operate.  I am curious as to how 'easily/fruitfully' these problems are optimized over vs. more traditional parameterised model searches?

---

### Decision · Program_Chairs · 2018-03-20
**ICLR 2018 Workshop Acceptance Decision**

**Decision:**

Accept

**Comment:**

Congratulations, your paper was accepted to the ICLR workshop.